# Surface Plasmon Resonance of Large-Size Ag Nanobars

**DOI:** 10.3390/mi13040638

**Published:** 2022-04-18

**Authors:** Fan Wu, Lin Cheng, Wenhui Wang

**Affiliations:** 1School of Textile Science and Engineering, Xi’an Polytechnic University, Xi’an 710048, China; fan.wu@xpu.edu.cn; 2Key Laboratory of Functional Textile Material and Product (Ministry of Education), Xi’an Polytechnic University, Xi’an 710048, China; 3Department of Physics and Max Planck Centre for Extreme and Quantum Photonics, University of Ottawa, Ottawa, ON K1N 6N5, Canada; lchen251@uottawa.ca; 4Ministry of Education Key Laboratory for Nonequilibrium Synthesis and Modulation of Condensed Matter, School of Physics, Xi’an Jiaotong University, Xi’an 710049, China; 5Shaanxi Province Key Laboratory of Quantum Information and Quantum Optoelectronic Devices, School of Physics, Xi’an Jiaotong University, Xi’an 710049, China

**Keywords:** large-size Ag nanobar, localized surface plasmon resonance, scattering cross-section, FDTD simulation

## Abstract

Silver nanobars have attracted much attention due to their distinctive localized surface plasmon resonance (LSPR) in the visible and near-infrared regions. In this work, large-size Ag nanobars (length: 400~1360 nm) working at a longer-wavelength near-infrared range (>1000 nm) have been synthesized. By using the finite-difference time-domain (FDTD) simulation, the LSPR properties of a single large-size Ag nanobar are systematically investigated. The LSPR in Ag nanobar can be flexibly tuned in a wide wavelength range (400~2000 nm) by changing the bar length or etching the bar in the length direction. Our work provides a flexible way to fabricate nanoparticle arrays using large-size nanobars and throws light on the applications of large-size nanomaterials on wide spectral absorbers, LSPR-based sensors and nanofilters.

## 1. Introduction

Due to the intense surface plasmon absorption and enhanced local electromagnetic fields [1,2,3], localized surface plasmon resonance (LSPR)-based metallic nanoparticles have been extensively exploited in solar cells [4,5], photocatalytic water splitting [6,7], and some promising label-free sensors [8,9]. The realization of multi-functionality strongly depends on the precisely controllable LSPR. Various metallic nanostructures synthesized by the “Bottom-Up” or “Up-Bottom” method have been utilized to adjust the plasmon resonance. Solution-grown nanostructures such as the cube [10,11], plate [12] and bar [13] show strong light absorption in the visible or near-infrared wavelength range. Metallic nanoparticle arrays and other complicated nanostructures with a flexible LSPR peak are fabricated by electron beam lithography (EBL) [14,15] or focused ion beam lithography (FIB) [16,17] for nonlinear optics [18,19] and ultrasensitive biosensors [20,21].

Among these nanostructures, silver (Ag) nanobars attract much attention due to their relatively low energy loss and strong light absorption in the near-infrared range [22,23]. In previous works, Ag nanobars with a length of below 200 nm have been widely synthesized to satisfy specific operation wavelengths (<1000 nm) for surface enhanced Raman scattering (SERS) sensing and catalyst applications [24,25]. The synthesis of large-size Ag nanobar (length > 200 nm) working at a longer wavelength (>1000 nm) is seldom reported.

Nanoparticle arrays are fabricated to enhance near field and reduce spectral linewidth. Generally, there are two ways to fabricate nanoparticle arrays: etching an Ag deposited layer by lithography technology [26] or regulating the spatial distribution of nanoparticle by the self-assembly method [27]. The deposition techniques are complicated, expensive, and require advanced instruments. In particular, the surface of the Ag deposited layer is coarse. Ag nanobars with smooth surfaces and controllable size can be synthesized by the polyol method in a low-cost and mass-produced way. Large-size Ag nanobars can be flexibly etched into various shapes to fabricate perfect nanoparticle arrays under the combination of the chemical synthesis method and lithography technology, thus leading to the broad-band tunable plasmon resonance, such as nanograting for label-free sensitive sensing [28], plasmonic oligomers for directional transmission of optical information at the nanometer scale [29], and even some complex coupled nanostructure arrays which could be used to develop dynamically reconfigurable metamaterials for promising applications in nanoparticle trapping and optical filters [30,31].

In this work, large-size Ag nanobars (length (*l*): 400~1360 nm, width (*w*) and thickness (*t*): 100–200 nm) working at the longer-wavelength near-infrared range are fabricated. We demonstrate the LSPR peak location of a large-size Ag nanobar can be flexibly tuned in a broad spectral range by changing the bar size or etching the bar into various shapes. Through systematically investigating the LSPR properties of large-size Ag nanobars using finite-difference time-domain (FDTD) simulation, we find that the resonance wavelength flexibly extends across 400 nm to 2 μm. We also study the sensitivity of a large-size Ag nanobar on the refractive index (RI) of ambient medium, as well as the polyvinylpyrrolidone (PVP) coating generally existing on the nanobar surface. The preparation of large-size Ag nanobars not only further supplements the near-infrared nanomaterial database, but also could provide a flexible solution of nano-arrays fabrication for high-performance sensors, absorbers and nanofilters.

## 2. Materials and Methods

Large-size Ag nanobars were synthesized by the polyol method using bromide ions. In details, 30 μL of 13.61 mM sodium bromide (NaBr, ≥99.0%, ethylene glycol (EG, analytical grade) as the solvent) was quickly dropped to 5 mL of EG solution heated at 160 °C under vigorous stirring conditions. After 30 min, 4.875 mL of EG solutions containing 93.99 mM silver nitrate (AgNO_3_, ≥99.8%) and 4.875 mL of EG solutions containing 144 mM poly (vinyl pyrrolidone) (PVP, M.W. = 40,000) and 0.144 mM sodium bromide (NaBr, ≥99.0%) were simultaneously dropped into the above solution at 0.375 mL/min using a syringe pump (LSP02-1B, Longer Pump). The chemical reaction took place for 1 h in a dark place. Then, 4 mL of acetone was added into 1 mL of the final product, then the solution was mixed with ultrasonic oscillation (ultrasonic frequency: 40 KHz and time: 5 min). After that, Ag nanoparticles were separated from the chemical reagents using the centrifugation method (10,000 rpm/min, 10 min). Then, the solution was removed (most Ag nanoparticles were attached to the inner wall of centrifuge tube) and the above process was repeated using ethanol and deionized water, respectively, instead of acetone. Finally, Ag nanoparticles were dispersed in ethanol for future use.

The sample for scanning electron microscope (SEM) study was prepared by drying some drops of the aqueous suspension of nanobars on silicon wafer under ambient conditions. SEM images were taken on an FEI field-emission microscope (JEOL JSM-7000F) operated at an accelerating voltage of 15 kV. The extinction spectrum of ethanol and Ag nanobars dispersed in ethanol were simultaneously taken at room temperature on a UV/VIS/NIR spectrometer (Lambda 750S, PerkinElmer). Note that ethanol was used as a reference solution to eliminate the effect of reagents on the absorption of incident light. The extinction spectrum in this work was obtained from Ag nanobars. The measurement of the extinction spectrum lasted for ~6.5 min, and the scanning speed was 266.75 nm/min.

The finite-difference time-domain (FDTD) simulation (Lumerical Solutions 2020) is applied to obtain the optical spectra and electric field distributions. The simulations are carried out in three dimensional space. The incident light (total-field scattered-field source) with a wavelength of 300–3000 nm vertically irradiates on the surface of nanostructures along the z direction with transverse-magnetic (^TM^) polarization. The optical properties of Ag are obtained from Palik [32]. A mesh size of 2 nm and a boundary of perfectly matched layers (PML, type: stretched coordinate PML, profile: standard, the number of layers: 8) are found accurate enough for the simulations. All the structure parameters are chosen according to the experiment sample.

## 3. Results and Discussion

Figure 1a shows large-size Ag nanobars and other small Ag nanoparticles are fabricated in the polyol synthesis. The high-magnification SEM image of Ag nanobars are displayed in Figure 1b. We can see the surfaces of large-size Ag nanobars are relatively smooth with very few observed defects, as shown in Figure 1b. As demonstrated in the length distribution map in Figure 1c, the length of the Ag nanobar is 400~1360 nm.

The optical extinction spectrum of the samples containing various sizes of Ag nanobars and other Ag nanoparticles is displayed in Figure 2a. The peak located near 435 nm with broaden extinction intensities is mainly induced by the transverse resonance mode of large-size Ag nanobars and the resonance of other small Ag nanoparticles coexisting in the products. The peak originated from the longitudinal mode of the large-size Ag nanobar appears in the near-infrared wavelength range, which is obviously different from the one with the narrower size distribution [25]. Two extinction peaks in the near-infrared (NIR) wavelength range can be attributed to two kinds of nanobar aggregation formed in the extinction measurements which show different NIR responses. The longer operation wavelength (>1000 nm) of the large-size Ag nanobar widens the working wavelength of existing NIR sensing. The extinction spectrum arises from the contributions of absorption and scattering of the samples. For large nanoparticles, the extinction intensity is mainly related to the scattering of light. The differential scattering cross section relates the intensity on a single particle to the power scattered by it per solid angle [33]. The simulated scattering cross-section of a single large-size Ag nanobar is shown in Figure 2b. The optical responses of the structure are determined by the polarization of the input laser beam [34,35]. Under experimental conditions, the nanobars would be randomly oriented in the solution. Throughout simulations in this article, we set the light polarization direction parallel to the nanobar length and only multiple longitudinal plasmon modes of nanobar are excited in the spectral range of interest. The simulation results represent a specific case of the possible arrangements, which brings us a clear physical mechanism about the plasmon behavior. The LSPR peak of the Ag nanobar located at the near-infrared wavelength region shifts to red with increasing nanobar length. For a nanobar with a length of 1400 nm (close to the maximum length in the experiments), the dipole resonance appears at ~1800 nm, which is different to the results shown in Figure 2a. This discrepancy can be attributed to the coupling among nanobar aggregates in the extinction measurements that lead to the red-shifting resonant wavelength. The corresponding local electric field distribution of a single large-size nanobar (*l* = 1100 nm and 1550 nm, respectively) excited by the resonance wavelength (λ_res_) is shown in Figure 2c. We can see that for the dipole resonance wavelength (see (i) and (iii)), the electric field is strongly confined in the corner and side of the nanobar, resulting in the increased scattering cross-section as displayed in Figure 2b. Moreover, higher order longitudinal modes are excited in the larger Ag nanobars with a length of 1550 nm (see (v)). The electric field distributions at resonant wavelengths are quite different due to the different surface charge distributions on Ag nanobars [36,37,38,39]. The SPRs in the visible and near-infrared region arise from the phase difference at nanobar ends and Fabry–Perot resonances of the surface guided waves occurring between the two ends of an individual Ag nanobar, respectively [40]. The red-shift of plasmon resonance wavelength with increasing nanobar length arises from the electromagnetic retardation effect [41,42].

To illustrate the size influence on the surface plasmon resonance of Ag nanobars, a series of simulations were performed on Ag nanobars with different length (*l*), width (*w*) and thickness (*t*). Figure 3a shows the simulated scattering cross-section of a single Ag nanobar with length varying from 300 nm to 900 nm. With increasing nanobar length, the dipole resonance is red-shifted from 465 nm to 1.2 μm, which is a broad-band range across the visible and near-infrared region. This can be attributed to the retardation effect caused by the increased nanobar length. High-order plasmon modes excited in the large-size Ag nanobars emerge in the visible light region. For a single Ag nanobar (*l* = 300 nm), the quadrupole plasmon mode appears at ~410 nm. When the nanobar length is above 300 nm, higher-order modes are excited. Due to the large red-shift of the dipole resonance compared with that of high-order resonance, the interaction between the dipole mode and high-order mode is weak in a large-size Ag nanobar [43]. In addition, the effect of width and thickness of the Ag nanobar (*l* = 600 nm) on longitudinal resonance modes have been investigated, as shown in Figure 3b. We can see with increasing nanobar width and thickness, quadrupole resonance at ~600 nm becomes distinguished as the charges decouple, while high-order resonance peak position (around 400 nm) is affected due to a small extent. The red-shifting and broadening quadrupole resonance peak is possibly due to the retardation and radiative damping effects. For the longitudinal dipole resonance, the peak location has almost no change when the nanobar length is kept constant. We thus infer that etching a large-size nanobar along nanobar length direction can broadly tune the LSPR peak.

Figure 4 shows the scattering cross-section of large-size nanobar (*w* = *t* = 200 nm, *l* = 700 nm) before and after the etch process. Three kinds of nanobar arrays are made by etching a large-size Ag nanobar along the length direction. Compared with the LSPR peaks in an Ag nanobar without being etched, as shown in Figure 4a, we find that the resonance wavelength has an obvious change. With an increasing number of etch times, all LSPR peaks appear at a shorter wavelength. The dipole resonance peak shifts from ~970 nm to ~450 nm by the etch process. Moreover, the scattering intensity is decreased by increasing the etch times. We simulate the scattering cross-section of an individual small bar (*l* = 25 nm, 50 nm, 125 nm and 150 nm, respectively) and find that the LSPR peaks appear at the visible wavelength region. When two or more small bars are placed side by side (the gap between the nanobars is 50 nm), the bonding dipole–dipole mode shifts to a longer wavelength and a new resonant peak is excited by the coupling effect of these small bars. The scattering cross-section of side-by-side nanobars far exceeds that of an individual small bar, which can be attributed to the coupling among bars resulting in the increased scattering efficiency of radiative modes [44]. In addition, the strong radiative damping introduced by in-phase oscillation of the individual small bar dipoles leads to the broad bandwidth in etched nanobars [45]. Consequently, we conclude the shift of surface plasmon resonance wavelength and change of intensity can be attributed to the coupling of plasmon modes excited in individual small bars, as the etched nanobar can be seen as the arrangement of side-by-side small nanobars. For an Ag nanobar with a larger length, the resonance wavelength is possible to be widely tuned from the visible to the near-infrared region by advanced etch technology. When the gap of side-by-side nanobars is decreased to several nanometers, the coupling effect of plasmons can enhance by generating gap plasmon resonance among small nanobars. The gap plasmon resonance modes, generally with a narrower full width at half maximum (FWHM) value, leads to a relatively high sensitivity and good figure of merit (FOM). The etch technology provides an effective approach to flexibly control the LSPR response of the large-size Ag nanobar for potential applications. It should be noted that the follow-up experiment is carrying on, and that through precisely controlling the morphology of etched nanobars, the tunable LSPR peak position change in a wide spectra range can be realized.

The RI sensing property of a large-size Ag nanobar is investigated in Figure 5. We can see from Figure 5a that the resonance wavelength of a large-size Ag nanobar red-shifts with the increase of the RI of ambient medium, which is consistent with the experimental and simulated results from other nanoparticle structures reported previously [29,46,47]. The maximum resonance wavelength and corresponding fitted curves for different RI are shown in the inset of Figure 5a. The slope of the line by definition Δλ_res_/Δn is equal to the sensitivity (S). The FOM can be defined as FOM = S/FWHM. The RI sensitivity of Ag nanobar (*w* = *t* = 200 nm, *l* = 900 nm) is calculated as 1193 nm/RIU (RIU is refractive index unit), and FOM is of 3.4 RIU^−1^, which is comparable with that of other single metal nanoparticle for ultrasensitive nanosensors [48]. The comparison of the LSPR performance for individual nanoparticles, clusters and arrays is shown in Table 1. The broad linewidth in the scattering cross-section of a single large-size Ag nanobar limits the FOM. LSPR-based nanosensors with high sensitivity and FOM are expected to be obtained by precisely designing the nanostructures by directly etching a single large-size nanobar into small-area arrays, or self-assemble a nanobar to make large-area arrays through the lithography process. Generally, in the polyol fabrication process with PVP as a surface active agent, the surface of the Ag nanobar is coated with a thin PVP layer (thickness < 5 nm). A thin PVP layer could vary the RI surrounding nanobar. To check the influence of the PVP layer on the LSPR property of a large-size Ag nanobar, we simulated the scattering cross-section of a single nanobar with different thickness of PVP layer, as shown in Figure 5b. It can be seen that the LSPR peak red-shifts with increasing thickness of the PVP layer, which can be attributed to the low oscillation energy experienced by localized surface plasmon in an Ag nanobar with a thicker PVP layer (RI of 1.5), and thus leading to the lower resonance frequency [37,49]. For thinner PVP coating (thickness: 3 nm), there is almost no obvious effect on the LSPR properties of a nanobar. That means that the effect of a thin PVP layer on Ag nanobars synthesized by the polyol method generally can be ignored.

## 4. Conclusions

Large-size Ag nanobars (length: 400~1360 nm, width and thickness: 100~200 nm) working at longer wavelength (> 1000 nm) have been synthesized through a polyol process method using bromide ions. Large-size Ag nanobars show excellent tunable LSPR from visible to infrared region (400 nm~2 μm) by changing the bar size or etching bar in the length direction. By systematically studying large-size Ag nanobars through FDTD simulation, we found that the dipole resonance peak is mainly related to the length of a nanobar. Through systematic investigation of an individual nanobar, we find the plasmonic properties of etched nanobars result from the coupling effect of arranged side-by-side small nanobars. In addition, through changing the refractive index of medium around nanobars, as well as the PVP coating thickness, the plasmon resonance of a large-size Ag nanobar can also be tuned. The simulated RI sensitivity of an individual large-size Ag nanobar can reach 1193 nm/RIU, and a FOM of 3.4. This work is expected to promote the development of large-size plasmon-based nanomaterials and their applications on the fabrication of nanoparticle arrays for broad-band absorbers, sensors and nanofilters.

## Figures and Tables

**Figure 1 micromachines-13-00638-f001:**
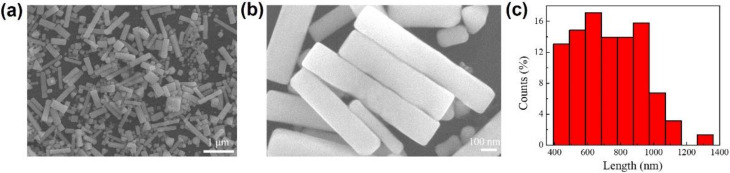
The characterization of large-size Ag nanobar. (**a**) Scanning electron microscope (SEM) image of Ag nanobars. (**b**) The high-magnification SEM image of Ag nanobars. (**c**) The length distribution map of Ag nanobars.

**Figure 2 micromachines-13-00638-f002:**
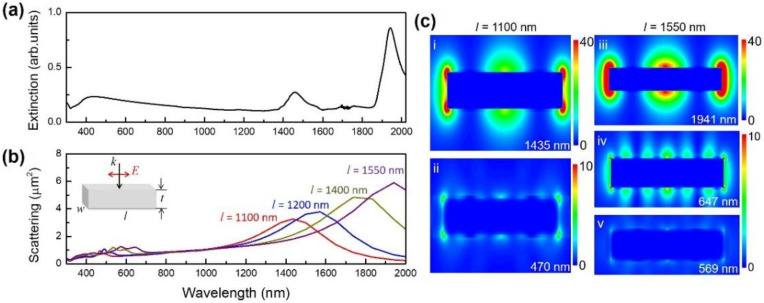
The localized surface plasmon resonance (LSPR) properties of a large-size Ag nanobar. (**a**) The experimental extinction spectrum taken from aqueous suspension of silver nanobars with various sizes. (**b**) The simulated scattering cross-section of two large-size silver nanobars (*w* = *t* = 200 nm, *l* = 1100 nm, 1200 nm, 1400 nm and 1550 nm, respectively) at an incident angle (*θ* = 0°) in air (RI = 1.0). (**c**) The corresponding electric field distributions for a nanobar (*w* = *t* = 200 nm) with two different lengths at on-resonance wavelengths (i) *l* = 1100 nm, λ_res_ = 1435 nm, (ii) *l* = 1100 nm, λ_res_ = 470 nm, (iii) *l* = 1550 nm, λ_res_ = 1941 nm, (iv) *l* = 1550 nm, λ_res_ = 647 nm, and (v) *l* = 1550 nm, λ_res_ = 569 nm.

**Figure 3 micromachines-13-00638-f003:**
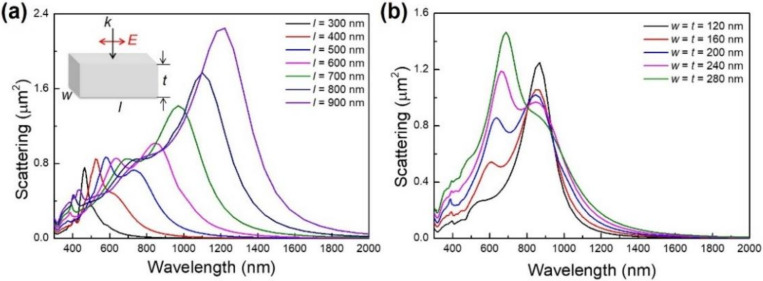
Finite-difference time-domain (FDTD) simulated scattering cross-section of single silver nanobar with various sizes at an incident angle (*θ* = 0°) in air (RI = 1.0). (**a**) Width and thickness (*w* = *t* = 200 nm) and lengths (*l* = 300 nm, 400 nm, 500 nm, 600 nm, 700 nm, 800 nm and 900 nm, respectively). (**b**) Length (*l* = 600 nm), width and thickness (*w* = *t* = 120 nm, 160 nm, 200 nm, 240 nm and 280 nm, respectively).

**Figure 4 micromachines-13-00638-f004:**
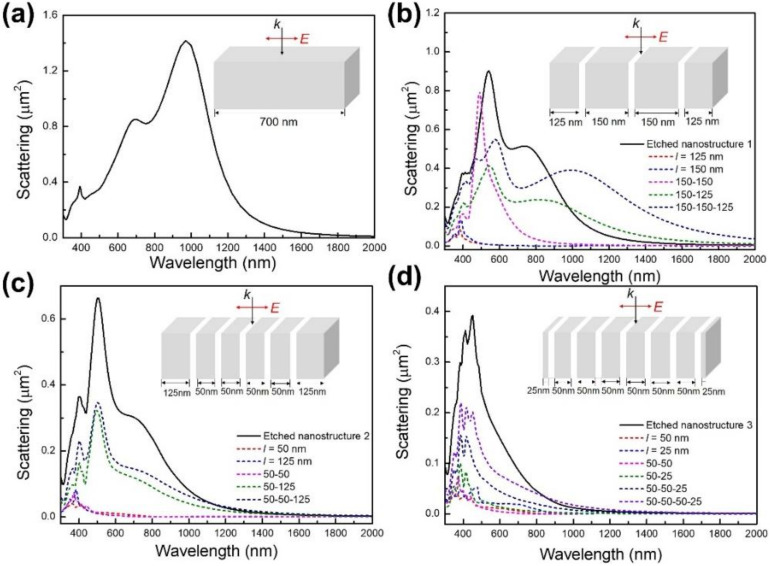
The LSPR property of a large-size Ag nanobar (*w* = *t* = 200 nm, *l* = 700 nm) before and after being etched along the length direction. (**a**) The simulated scattering cross-section of an original large-size Ag nanobar and (**b**–**d**) three etched nanostructures (black solid lines), individual nanobars (red and blue dashed lines) and the coupling effect of small-size nanobars (other colored dashed lines) simulated at an incident angle (*θ* = 0°) in air (RI = 1.0). The insets show the corresponding schematic diagrams of a large-size Ag nanobar before and after the etch process. The width and thickness of each etch in white is 200 nm and the length is 50 nm.

**Figure 5 micromachines-13-00638-f005:**
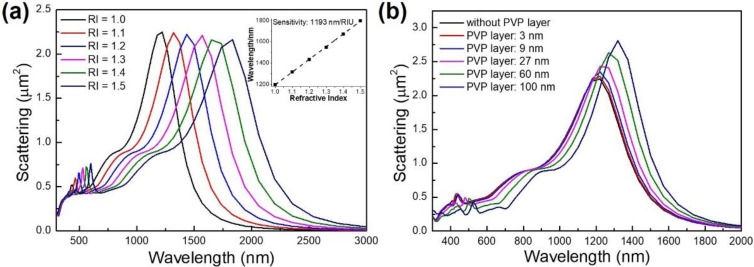
The influence of ambient medium and polyvinylpyrrolidone (PVP) coating on the surface plasmon resonance of a large-size Ag nanobar. (**a**) The simulated scattering cross-section of a single silver nanobar (*l* = 900 nm, *w* = *t* = 200 nm) in different ambient medium with various refractive indexes (RI = 1.0, 1.1, 1.2, 1.3, 1.4, 1.5) at an incident angle (*θ* = 0°). The inset shows the relationships between maximum resonance wavelength with the refractive index of the surrounding medium. (**b**) The simulated scattering cross-section of a single silver nanobar (*l* = 900 nm, *w* = *t* = 200 nm) with different thicknesses of PVP coating. The thickness of the PVP layer is 3 nm, 9 nm, 27 nm, 60 nm and 100 nm, respectively.

**Table 1 micromachines-13-00638-t001:** LSPR sensitivities for nanostructures used in the experiments and simulations.

Nanostructure	Sensitivity (nm/RIU)	FOM (RIU^−1^)	Type	Reference
Rodlike Ag nanoparticle	235	4.1	experiment	[46]
Ag nanobar	1193	3.4	simulation	this work
Ag nanocube	1565	5.4	experiment	[47]
Au bipyramids	2000	4.5	experiment	[50]
Au nanodisk heptamer	2340	5.4	experiment	[51]
Au heptamer	940	20.9	simulation	[29]
Ag sphere septamer	515	10.6	simulation	[45]
Au nanobar array	600	4.68	simulation	[52]
Au nanodisk array	853	126	simulation	[20]
Cross-hair/nanorod combination	1200	26.67	simulation	[36]
Ag nanorod arrays with the connected veins	800	12.17	simulation	[38]

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
