# Peer review of "Surface Plasmon Resonance of Large-Size Ag Nanobars"

_micromachines, 2022, doi:10.3390/mi13040638_

Round 1

Reviewer 1 Report

The paper has been improved. But a few of the issues I previously mentioned hasn’t been properly addressed in the revised manuscript. I suggest the following modifications:

1. Regarding the fabrication process, the authors have discussed this well in the review response letter. However, the details mentioned in the letter are not present in the revised manuscript. Please include all the details in the manuscript as well (the paragraph in the review response letter starting with “4mL of acetone was added…”).

2. In the manuscript, please mention that “Nano Measurer” software was used on the SEM images. Again, this detail was mentioned in the review response letter, but not in the revised manuscript.

3. The explanation regarding why there are two peaks in Figure 2(a) is not very clear. It is not obvious why an aggregate of Ag nanobars would produce two peaks. If you note Figure 2(b), it is obvious that different sized nanobars would produce peaks at different wavelengths. If there are a range of Ag nanobars in the sample with length ranging from 1100nm to 1550nm (for example), then you would expect peaks for each length starting from lambda = 1400nm to lambda = 2000nm. If there are lot of sizes in this range, then there would be a lot of peaks. All of these peaks would overlap to create a very wide peak extending from lambda = 1400nm to lambda = 2000nm. An aggregate of nanorods should produce a plot like that. However, Figure 2(a) is completely different from this. The simulation does not need to perfectly match with the experimental data. However, the experimental and simulation data should be consistent with each other. Currently, this is not the case. This should be addressed before the paper can be considered for publication.

The current explanation regarding aggregate of nanobars is not backed up by the simulation data. Perhaps random orientation of the nanobars may have something to do with this. But any behavior related to aggregate of nanobars with different lengths and orientation should still be a smeared out peak instead of two distinct peaks.

4. Regarding the reference section, the authors may wish to include Prof. Lambertus Hesselink’s paper on plasmonic Archimedean spirals that show polarization dependent response of spiral structures.

5. Regarding my comment 10, the authors have indeed mentioned in the revised manuscript that the electric field direction was assumed to be parallel to the nanobar length. However, they have not mentioned in the paper that under experimental condition, the nanobars would be randomly oriented. Thus, the simulation results only show a specific case of the possible arrangements.

I would recommend the authors to go through my previous comments more carefully and to revise the paper accordingly.

Reviewer 2 Report

In this work, the authors synthesized a large size of Ag nanobars working at a longer-wavelength near-infrared range. They claimed the sensitivity could reach 1193 nm/RIU. However, the following points should be addressed before accepting this manuscript (with major revision). 

  1. The extinction and cross-section scattering should be defined in the text for experiment and simulation.
  2. The version of Lumerical’s FDTD solver should be mentioned in the text. Besides, the incident light pattern (e.g., plane wave, pulse, or Gaussian wave) and the location and size of PML should be mentioned in the text.
  3. It is evident in Fig. 2(c) that the different standing wave numbers of surface plasmon waves occurred on the surface of Ag nanobar varying with different lengths of Ag nanobar. However, the authors didn’t give an explanation on this point. Please clarify the mechanism (see Plasmonics, 2017, 12, 277-285) and cite the related reference.
  4. Figure 4 shows the scattering cross-section of large-size nanobar before and after the etch process. Is Fig. 4 the experimental or simulated result? Please briefly clarify what happens if the gap of side-by-side nanobars is changed.
  5. It states that, “It can be seen that the LSPR peak red-shifts …, which can be attributed to the … and thus leading to the lower resonance frequency.” To support this description and the mechanism of this result, the related reference (e.g., Nanoscale Res. Lett., 2016, 11, 411, and J. Opt. 18 (2016) 115003) should be cited in the text.
  6. Is Fig. 5 the experimental or simulated result? How can you prepare the medium under test (MUT) with n=1.1 and 1.2 if Fig. 5 is an experimental or simulated result?

7. The figure of merit (FOM) is also an import performance of sensitivity. Please calculate the corresponding FOM of the synthesized structure in Fig. 5(a).

  1. The references used in the “introduction section” are not enough. 

(a)Many plasmonic sensing structures with Ag nanorods have been published with better sensitivity and frequency responses. To be beneficial for the readers to know the other approaches, the manuscript should include the different designs with the current sensing structure, e.g., Nanomaterials (2020), 10, 493 and Results in Physics, 2019, 15, 102567). 

(b)The positive-negative charge density diagram can help the readers to understand the mechanism of LSPR in the metal nanorod structures. In this manuscript, the mechanism of the LSPR arising from Ag nanobars should be improved and quote the related literature. Several reported articles have addressed the simulated scattering cross-section (e.g., Applied Opt. 2009, 48, 617-622, and J. Phys. D: Appl. Phys. 50 (2017) 125302) and positive-negative charge density diagram (J. Appl. Phys. 2016, 120, 093110 and Appl. Opt. 58, 2530-2539(2019)) of silver nanobars in detail.

(c)   Authors claimed the sensitivity of this work could reach 1193 nm/RIU. I suggest providing a comparison table regarding the sensor performance between this work and reported articles.   

Reviewer 3 Report

The paper investigated the surface plasmon resonance (SPR) of large size Ag nanorods.

  1. As far as I know, large size Ag nanorods investigated in this study are commercially available and the SPR of such Ag nanorods have been already well studied. Therefore, the novelty of this paper is not very clear.
  2. In Fig. 4, the effect of multi-size Ag nanorods was investigated. However, the calculation model seems to have quasi-periodic, but actual samples are random configuration. Please add discussion the effect of the configuration.
  3. Please add more discussion about applications.

Round 2

Reviewer 1 Report

The paper has been improved. It can be considered for publication now.

Reviewer 2 Report

The authors have revised their manuscript according to my comments. This manuscript can now be accepted for publication.

Reviewer 3 Report

The authors addressed my comments.